# The $R = 1$ threshold can misclassify epidemic stability
Kris V. Parag [1,2] ✉, Mauricio Santillana[3,4,5], Anne Cori[2] & Uri Obolski[6]

The effective reproduction number, $R$, is a predominant statistic for tracking infectious disease spread and informing health policies. An estimated $R = 1$ is universally interpreted as a stability threshold distinguishing epidemic growth ($R > 1$) from control ($R < 1$). We demonstrate that this interpretation frequently fails because $R$ typically averages over groups with heterogeneous characteristics. We find that $R = 1$ conceals valuable early-warning signals of resurgence and misclassifies complex dynamics as noise, generating false positive stability thresholds that diminish predictive and policymaking value. We further illustrate that a popular alternative transmissibility definition (using next-generation matrices) overcorrects this issue, producing false negative stability signals by amplifying stochastic variation. We address these limitations by adapting a recently developed statistic, $E$, derived from $R$ using experimental design theory. We show that $E$ tightly constrains the set of scenarios consistent with stability, while remaining robust to noise and establish $E = 1$ as a more practical and meaningful real-time threshold.

Accurately tracking the transmissibility of infectious diseases is a long-standing and important problem. Timely indicators of the growth or decline of epidemics contribute valuable evidence for informing public health policy, assessing interventions, improving epidemic response and supporting pandemic preparedness[1,2]. The *effective reproduction number*, $R$, is the most popular statistic for describing transmissibility. $R$ estimates the ratio of expected new infections to actively circulating (past) infections[3,4]. Although there are many different approaches for computing $R$[1,5], its role as a key threshold statistic is a fixture across epidemiology. An estimated $R = 1$ indicates that the incidence of new infections will remain roughly constant or stable, while values above or below this threshold foretell of rising or falling infection incidence.

This interpretation is ubiquitous and its simplicity underlies why $R$ is prominent as a predictive statistic for guiding public health responses and communicating the state of an epidemic[6]. However, as $R$ is typically estimated from incidence data at coarse spatial scales (nationally or regionally)[1,7], it frequently averages over groups with heterogeneous dynamics. As a consequence, finer scale variations that may contain critical signals for prediction and control are neglected. Group heterogeneities emerge from variations in behaviour, sociodemographic factors, immunity levels, location and other characteristic features[8,9]. Such heterogeneity-driven transmission patterns were frequently observed during the recent COVID-19 pandemic and across recurring Ebola virus, SARS, influenza and and other outbreaks[9–14].

Many attempts have been made to refine effective reproduction numbers, both in formulation and estimation. These approaches generally incorporate known heterogeneities to limit biases in estimates or improve the construction of epidemic stability thresholds[8,15–19]. While these developments are valuable, they commonly require auxiliary data such as contact matrices or mobility indices (which can induce other biases due to unexpected time-variations[20,21]) or rely on computationally complex models. This limits their use for real-time outbreak tracking and in locations lacking sophisticated surveillance. Effectively balancing practical surveillance and computational constraints against the need to capture population heterogeneity remains a challenge to producing meaningful estimates of transmissibility in real time.

Here we provide insights into this challenge for real-time applications and offer solutions that require no auxiliary data and reflect federated surveillance settings[22]. We achieve this by questioning the universal role of the $R = 1$ threshold and exposing its limitations for indicating epidemic stability in heterogeneous settings. Using theory, simulations and empirical data, we highlight how averaging over groups, as employed by predominant formulations of $R$, mean that $R = 1$ seldom signifies stability even under very mild assumptions. We further show that improving the formulation of the

[1]Department of Engineering, King's College London, London, UK. [2]MRC Centre for Global Infectious Disease Analysis, Imperial College London, London, UK. [3]Machine Intelligence Group for the betterment of Health and the Environment, Network Science Institute, Northeastern University, Boston, MA, USA. [4]Department of Epidemiology, Harvard T.H. Chan School of Public Health, Cambridge, MA, USA. [5]Network Science Institute, Northeastern University, Boston, MA, USA. [6]School of Public Health, Department of Epidemiology and Preventive Medicine, Faculty of Medical and Health Sciences, Tel Aviv University, Tel Aviv, Israel. ✉e-mail: k.parag@imperial.ac.uk

**Fig. 1 | Space of $R = 1$ and $E = 1$ solutions for varying group reproduction numbers $R_j$. a** For $p = 2$ groups, we consider $R_1$ (x-axis) and its weight $w_1$ (y-axis). Weights always sum to 1 so $w_2 = 1 - w_1$. There are an infinite number of solutions yielding $R = 1$. We sketch the subset of those solutions (blue lines with dots at the end) satisfying the constraint $\sum_j R_j = p$, which means the $R_j$ have an arithmetic mean of 1. One solution line sets $R_1 = R_2 = 1$ for all weights and the other sets $w_1 = w_2 = \frac{1}{2}$ for all values of $0 \leq R_1 \leq 2$. This line includes many scenarios in which $R = 1$ hides a resurging group ($R_j > 1$). In contrast, $E = 1$ has a unique solution (red dot) guaranteeing $R_1 = R_2 = 1$ (also max $R_j = 1$). **b** We plot global statistics (y-axis) that we term $X$ (see later Eq. (3)) for the subset of the $w_1 = w_2$ solution line from (**a**) over which $R_1$ is resurgent. $R$ (blue) is unresponsive to the growing $R_1$, $E$ (red) indicates resurgence and max $R_j$ (black dashed) is the most sensitive. (**c**): for $p = 5$ groups we sample $R_j$ values (green histograms) from gamma distributions (see 'Methods' for simulation details). These use past incidence from times 1 to $t$, denoted as $I_1^t$. The mean of the samples is $E[R_j]$ (green dashed with maximum $E[R_j]$ in black) and we constrain the sum over groups to be 5 (the arithmetic mean of the $E[R_j]$ is 1). The subplots, from top left to bottom right, show scenarios with increasing $R_j$ heterogeneity among groups. We construct histograms for global statistics $X$ from the $R_j$ samples (reproduction numbers or $X$ are on the x-axis). Across the subplots, we find that an overall $R = 1$ (blue) conceals several resurging groups. $E$ (red) exposes these dynamics without being as sensitive as max $E[R_j]$.

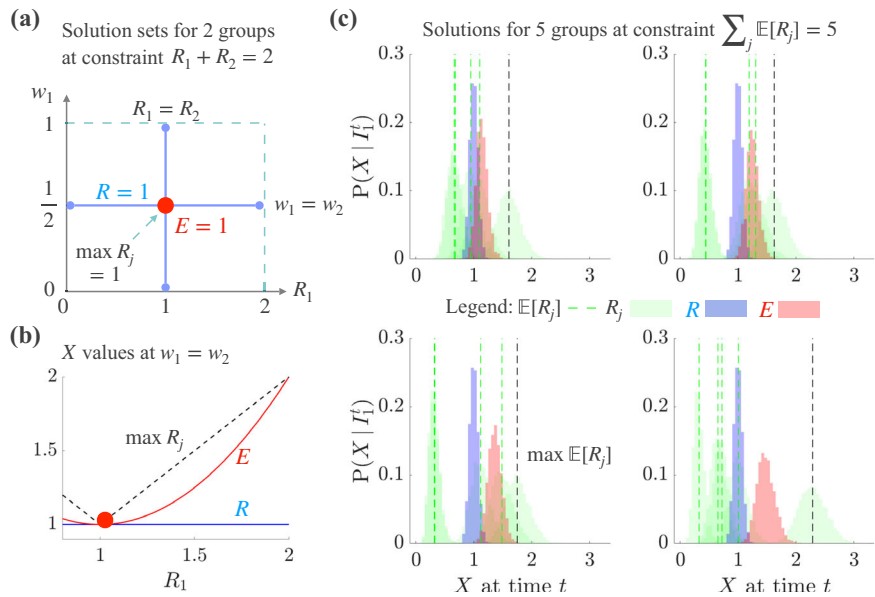

$R = 1$ threshold to incorporate heterogeneity as in popular next-generation matrix methods[23], causes extreme sensitivity to stochasticity within the groups because it essentially replaces averaging with a maximising operation. Consequently, $R$ may have substantial disadvantages as both a prospective and retrospective statistic[24], especially at the large scales over which it is most often inferred and reported. We offer a simple solution to these issues by embedding both the averaging and maximising operations inherent to different formulations of reproduction numbers in a common framework. We show that a recently introduced statistic, known as the risk averse reproduction number $E$[12], which was derived from the principles of statistical experimental design, sits within this framework and sensibly interpolates between these operations. As a result, we find that $E = 1$ is a more reliable and meaningful threshold for signifying epidemic stability in real time.

## Results
### Reproduction numbers may provide false positive indicators of stability

To interrogate the above claims, we consider $p > 1$ heterogeneous groups with homogeneous mixing assumed within each group. We use $R_j$ to denote the reproduction number of group $j$. This is the average number of secondary infections that a primary infection generates in that group. Using a renewal process framework (see 'Methods'), we can define the infectious force $\Lambda_j$ as the convolution of past infections in group $j$ (i.e. past $I_j$ values only) with the generation time distribution of that group. This distribution captures the random time intervals between primary and secondary infections[25]. We do not model interconnections among groups (e.g. travel or migrations) as we represent how infections are tracked during unfolding outbreaks across communities, where auxiliary time-dependent data that are needed to estimate these connections is likely unavailable, biased, or at best only snapshots in time are accessible[26].

The $R$ that is frequently computed at regional or national scales, based on the aggregate infections occurring in all groups[3,27,28], is equivalent to the weighted mean in Eq. (1) (see 'Methods' for the derivation of this equivalence from conventional formulae). The weights represent the proportion to the relative infectious force of each location or group, which seems an intuitive way to compose the overall transmissibility.

$$R \overset{\text{def}}{=} \sum_{j=1}^{p} w_j R_j, \text{ with weights } w_j = \frac{\Lambda_j}{\sum_{i=1}^{p} \Lambda_i}. \qquad (1)$$

However, by manipulating Eq. (1), we find that the $R = 1$ threshold condition is satisfied by any of the infinite number of solutions to $\sum_{j=1}^{p} \Lambda_j (R_j - 1) = 0$. Consequently, a vast and diverse space of epidemiologically important scenarios can all result in the same $R = 1$.

This simple observation has important ramifications. Even if we additionally constrain, as in Fig. 1, the mean of the group reproduction numbers to be 1 so that $p^{-1} \sum_{j=1}^{p} R_j = 1$, the $R = 1$ threshold still fails to distinguish among many divergent epidemic scenarios. At the simplest $p = 2$ setting, Fig. 1a, b demonstrates that these scenarios include numerous solutions with exponentially growing infection counts i.e. $R_1 > 1$. This discordance between the common interpretation of $R = 1$ and the underlying group dynamics becomes harder to diagnose as the number of groups $p$ becomes larger. This figure also includes another reproduction number $E$ that we will derive and discuss in a later section for comparison.

Figure 1c confirms that $R = 1$ can conceal diverse scenarios when we generalise Eq. (1) (see 'Methods' for local and global models behind the Fig. 1c simulations) to include the stochasticity of the infection counts observed in groups, in accordance with the most widely used estimators of $R$[3]. There we use X to generically refer to any estimator of transmissibility

(some of which are derived in subsequent sections), with blue for $X = R$ and green for every $X = R_j$. We examine the expectation $E[R_j]$ across samples from the group reproduction numbers (histograms reflect uncertainty arising from the stochasticity) and observe numerous settings in which multiple $E[R_j] > 1$ (and often no single $E[R_j] = 1$). Yet $R$ still confidently signals that transmission is stable ($E[R] = 1$ and the variance is small). We term this a false positive signal of overall epidemic stability because local instability (growth) occurs.

False positive signals of stability are misleading for two main reasons. First, they conceal the dynamics of groups experiencing resurging infections, which only become apparent when the infections in those groups grow to substantial levels[29]. As the effectiveness of interventions is known to be tightly coupled to how quickly they are implemented[30], this delay can limit overall epidemic response in impact and cost. Second, these false indicators of stability may inform premature release of existing controls or, when communicated to the public, engender risky behavioural responses. Both factors can accelerate undetected growth and resulting burden of the disease. These detrimental effects worsen under realistic surveillance, where incomplete reporting further obscures and delays the detection of resurgent dynamics[31].

## Next generation matrices may provide false negative indicators of stability

A counterpoint to the standard $R$ may be that, despite its frequent use in practice, it assumes homogeneity and is therefore not suited for assessing the stability of inherently heterogeneous systems. Next generation matrix and related methods[23,32] were proposed to rigorously compute epidemic thresholds for heterogeneous epidemic models. While these methods are mostly used to evaluate the basic reproduction number, $R_0$, of heterogeneous compartmental models, e.g. susceptible-infected-recovered (SIR) models with multiple infectious classes, we can still explore their relevance for our (renewal) modelling framework.

The eigenvalues of the next generation matrix (and associated approaches[8]) relate to the poles of the heterogeneous model[32]. Poles are central quantities for formally assessing the stability of linear dynamical systems[33] and are solutions to the denominator of the epidemic transfer function[34]. Transfer functions are standard tools that map how inputs (in our case imported infections) lead to outputs (the epidemic) and are computed in the Laplace domain as the ratio of outputs to inputs. We used transfer functions below and in Supplementary Note 2 to better link reproduction numbers to poles in a generalisable and interpretable manner, and to clarify relationships between local and global dynamics (which are fully described with poles). We sketch the transfer function construction below (see 'Methods' for a complete derivation, [33] for general transfer function theory and [34] for epidemic transfer functions).

The Laplace transform of the infections in group $j$ is $I_j(s) = \left(1 - R_{0j}G_j(s)\right)^{-1}M_j(s)$ where $s$ is a complex variable. $G_j(s)$ and $M_j(s)$ are the transforms for the generation time distribution and the imported infections for group $j$, respectively. The transfer function or ratio of output to input is $I_j(s)M_j(s)^{-1}$ with poles as the values of $s$ solving $R_{0j}G_j(s) = 1$. If we assume a seed infection starts the epidemic in all groups, then $M_j(s) = 1$ (the transform of a delta function). We neglect susceptible depletion effects so that $R_{0j}$ represents the basic reproduction number of group $j$. The stability of the epidemic is controlled by the dominant pole $s = r_j$, which is also the growth rate of group $j$. The other poles shape transient dynamics such as infection peaks and oscillations. At $r_j = 0$, critical stability occurs (infections plateau to a constant value) and $R_{0j} = 1$ as $G_j(0) = 1$ (this follows from the property of distributions summing to 1)[34].

Applying this construction, we obtain the Laplace transform of the total infections across the $p$ heterogeneous groups as the sum of $I_j(s)$ i.e. $I(s) = \sum_{j=1}^{p} I_j(s)$. This gives Eq. (2), which sums the transfer functions from all groups and reveals that the overall system poles are the union of

all the poles from every group. We let $r$ be the overall dominant pole of Eq. (2).

$$I(s) = \sum_{j=1}^{p} \frac{1}{1 - R_{0j}G_j(s)} \Longrightarrow r = 0 \Longleftrightarrow \max_j R_j = 1 \qquad (2)$$

If $r = 0$, the overall system is critically stable and the epidemic threshold is achieved. This is only possible when every $r_j \leq 0$ with at least one of the $r_j = 0$. If any $r_j > 0$ the overall system inherits this unstable group pole. We can guarantee that Eq. (2) does not become unstable by requiring $\max r_j = 0$, meaning that no $R_{0j} > 1$. As these reproduction numbers can be treated as piecewise-constant approximations of $R_j$[8], we get the condition in Eq. (2) that $\max R_j = 1$ is the correct threshold indicator for our deterministic, heterogeneous system. Next generation matrix approaches yield maximum eigenvalue conditions and those eigenvalues precisely map to epidemic poles. The $\max R_j = 1$ threshold guarantees that the epidemic cannot grow.

Returning to the simplest $p = 2$ setting of Fig. 1a, this maximum statistic shrinks the space of threshold solutions to one symmetrical point. This ensures that there are no false positive signals of stability as $\max R_j = 1$ requires every $R_j = 1$. However, this statistic displays maximum sensitivity to the group dynamics, as we see in Fig. 1b. Since real epidemics feature stochastic trajectories, we find in Fig. 1c that using a $\max E[R_j]$ epidemic threshold leads to frequent false negative indications of stability i.e. it signals instability even when groups are stable because the max operation magnifies random fluctuations. These overly conservative assessments can inform unnecessary deployment or maintenance of interventions and restrictions, leading to elevated economic and societal burden without counterbalancing benefits (although these assessments may work for very stringent policies aiming to fully suppress infections).

## Balancing sensitivity and averaging in reproduction number statistics

Although $R$ may average over group fluctuations to the degree that informative signals are lost, the $\max R_j$ threshold (from next generation matrix approaches) may instead amplify group fluctuations such that noise becomes misconstrued as signal. How can we moderate this issue? We want to sensibly leverage the information from the local $R_j$ to derive a global statistic $X$ such that $X = 1$ balances between averaging and upweighting local or group-level dynamics. Here, we propose a solution by recognising that both $R$ and $\max R_j$ are extremes within a continuum of weighted means. We construct these means in Eq. (3) with $\gamma \geq 0$ as a parameter that traverses this continuum. Equation (3) has no explicit infectious force terms, but these influence the estimates of $R_j$ (e.g. controlling its mean and variance) and hence the weights $\omega_j(\gamma)$.

$$X(\gamma) \stackrel{\text{def}}{=} \sum_{j=1}^{p} \omega_j(\gamma) R_j, \text{ with weights } \omega_j(\gamma) = \frac{R_j^{\gamma}}{\sum_{i=1}^{p} R_i^{\gamma}} \qquad (3)$$

Equation (3) defines a sequence of power means that provide baseline detection at small powers i.e. recover the smoother (baseline) trend from among the $R_j$. At $\gamma = 0$, we see that $X(0) = p^{-1}\sum_{j=1}^{p} R_j$ is the arithmetic mean, which matches $R$ in Eq. (1) when the groups have similar infectious forces. As $\gamma$ rises, Eq. (3) instead offers peak detection i.e. emphasises fluctuations. We observe this other extreme as $X(\infty) = \max R_j$. If we can optimise $\gamma$, we should be able to balance between the false positive stability signals that stem from over-averaging and the false negative ones from amplifying noise, to obtain a more reliable epidemic threshold. Note that if every group has identical transmissibility $R_j = \bar{R}$, then all the $X(\gamma) = \bar{R} = R = \max R_j$.

In earlier work from some of the authors, experimental design theory was used[12] to minimise the maximum uncertainty across all group level reproduction number estimates. This yielded the *risk-averse reproduction*

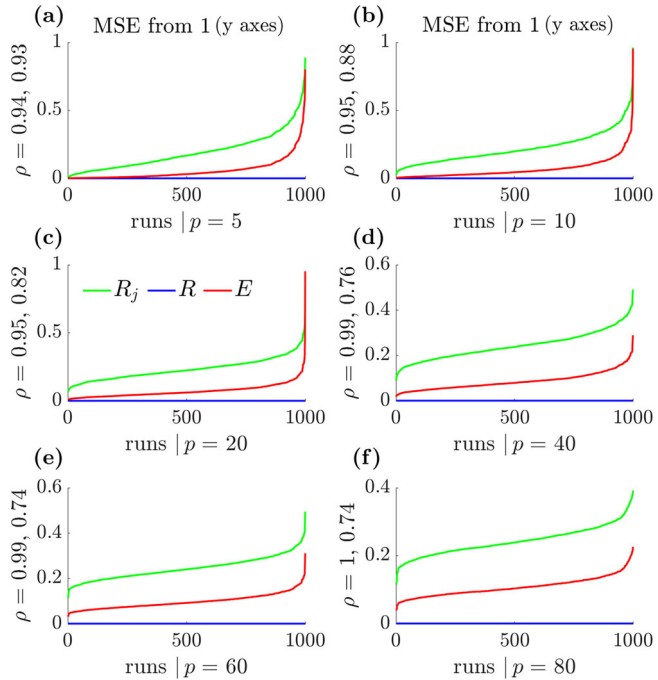

**(a)** MSE from 1 (y axes)  runs | $p = 5$

**(b)** MSE from 1 (y axes)  runs | $p = 10$

**(c)** $R_j$ — $R$ — $E$  runs | $p = 20$

**(d)** runs | $p = 40$

**(e)** runs | $p = 60$

**(f)** runs | $p = 80$

**Fig. 2 | Mean squared error (MSE) curves about 1 for $R$, $E$ and the groups $R_j$.** In (**a**–**f**), we simulate $R_j$ values that are consistent with $R \approx 1$ for differing numbers of groups $p$. We sample 1000 weightings, termed runs, from Dirichlet distributions. The runs set various levels of group heterogeneity and for each run we generate 5000 samples of every $R_j$ using the framework described in the 'Methods' and applied in Fig. 1c. The combined MSE of the $R_j$ across every run relative to 1 i.e. the mean of $\sum_{j=1}^{p} (R_j - 1)^2$ over the $R_j$ samples (green), is computed and ordered by how different the combined $R_j$ are from $R$. The MSE tends to 0 as all the $R_j$ get closer to $R$. We also calculate the means of $(R - 1)^2$ (blue) and $(E - 1)^2$ (red) under the above ordering to obtain global statistic MSEs about 1. We find that $E$ reflects the overall behaviour of the $R_j$ well, following the pattern of its MSE variation about 1, whereas $R$ ignores these local variations. We do not show $(\max R_j - 1)^2$ but note that while its curves do follow the pattern of the local combined $R_j$, they are substantially above those for $E$ (larger MSE) and are very noisy. We also present correlation coefficients $\rho$ between the combined MSE of $R_j$ and the MSEs of $E$ (first value) and $\max R_j$ (second value).

number $E$, which solved what is known as an E-optimal design to improve resurgence detection[35]. This statistic also sits within the framework developed here, as we find that $E = X(1)$ as shown in Eq. (4).

$$E \stackrel{\text{def}}{=} \sum_{j=1}^{p} \omega_j(1) R_j, \quad \text{with weights} \quad \omega_j(1) = \frac{R_j}{\sum_{i=1}^{p} R_i} \qquad (4)$$

Equation (4) weights how groups contribute to the overall measure of heterogeneous transmission using their relative transmissibility. This has similarities to $R$, replacing weighting by relative infectious potential in Eq. (1) with relative transmissibility. Equation (4) is also a softer approximation to $\max R_j$ and has the ingredients to achieve the balance we seek.

We therefore test if the new stability threshold signified by $E = 1$ is more meaningful than $R = 1$ and $\max R_j = 1$. First, we observe that $E = 1$ requires $\sum_{j=1}^{p} R_j (R_j - 1) = 0$, which collapses the space of solutions in Fig. 1a to the single point $R_1 = R_2 = 1$. This matches the $\max R_j$ threshold but is less sensitive to changes in group reproduction numbers, as we see in Fig. 1b. This is in contrast to the $R$ threshold, which remains at 1 throughout a wide range of scenarios with $R_1 > 1$. Moreover, $E$ has important advantages over $R$

and $\max R_j$ when we consider group-level stochasticity. In Fig. 1c, we validate these points at larger $p$, demonstrating that unlike $X(0)$ (constrained to 1) and $R$ (approximately 1), $E$ does not falsely signal stability. It also does not amplify noise like $X(\infty) = \max R_j$, which is always significantly above 1.

Figure 2 verifies that $E$ sensibly moderates how group variations inform overall stability. We test a range of weight vectors that result in $R \approx 1$. We fix the total infectious force $\Lambda$, set $\Lambda_j = p^{-1}\Lambda$ and then draw $p$ weights 1000 times from a symmetric Dirichlet distribution with parameter 4. These weight samples, which we term 'runs', set the fraction of total infections $I$ contributed by each group $I_j$ and hence control heterogeneity among the $R_j$. Setting $I = \Lambda$ ensures that overall $R = 1$. For each run, we sample 5000 posterior estimates of $R_j$ using the gamma distributions described in the 'Methods', which have mean $I_j \Lambda_j^{-1}$ in line with refs. 3,4 and 29. We examine the distance of all the sampled $R_j$ from 1 using their combined mean squared error (MSE) at every run. MSEs are ordered by how different the $R_j$ are from $R$.

We use those same runs and samples to construct $E$ as in Eq. (4) and get its MSE about 1. We observe that $E$ converges to 1 i.e. its MSE goes to 0, only when the overall MSE of the $R_j$ tends to 0 and hence when all $R_j$ converge towards 1. Consequently, $E$ provides a global measure of stability that reflects the patterns of stability among the local groups while considering their heterogeneity. We also compute (but do not show) the MSE of $\max R_j$ from the same samples. This statistic has a MSE that is much larger and noisier than $E$ but does still somewhat reflect the falling MSE trend among the $R_j$.

## Important signals that are occluded by common stability thresholds

Having proposed $E = 1$ as a more representative and balanced stability threshold, we simulate and enumerate important instances where these distinctions matter and can be occluded by routine use of available data. We simulate epidemics in two regions or groups ($p = 2$) using renewal models for each region. New infections are sampled from a Poisson distribution with mean that is the product of the local $R_j$ at that time and the convolution of past infections with the generation time distribution (see 'Methods'). We parametrise this distribution to reflect the dynamics of Ebola virus disease. In one deme the $R_j$ ground truth is sinusoidal with offset of 1.5 and amplitude of 1.4. The other deme computes its $R_j$ via a control algorithm that increases (by a factor of 1.1) or decreases (factor of 0.8) its $R_j$ at the last time step, based on whether the total incidence of both groups is below or above a desired sum. This creates scenarios that are critically stable globally despite time-varying transmissibility in both demes.

Figure 3a plots true incidence and the one-step-ahead estimates of that incidence, which are the best estimates of infections at the next time step given past incidence and estimates of $R_j$[36]. Figure 3b shows these estimates for each $R_j$, which minimise mean squared errors from the ground truth[28]. The global incidence is the sum across demes, and the one-step-ahead estimate of this and the associated $R$ follow from fitting the total incidence to a global Poisson renewal model. We estimate $E$ by sampling from our $R_j$ posterior estimate distributions and applying Eq. (4).

We construct these simulations to highlight important and different scenarios, A–C, which lead to an inferred global $R = 1$ in real time. The one-step-ahead incidence estimates also confirm overall stability. In A–C, $R$ is strongly confident in its assessment of stability (i.e. it has narrow credible intervals). However, the group-level (local) one-step-ahead estimates present diverse trends that are reflected by the time-varying $R_j$ estimates. The risk averse reproduction number $E$ (computed from these $R_j$ estimates) converges to the standard $R$ only when all groups are approximately stable. This occurs in scenario A. In contrast, B-C feature growing demes, indicating that $R$ presents false indicators of stability. Note that in all the scenarios $R$ features small credible intervals, whereas those of $E$ widen for B and C, where there is disagreement on whether the epidemic is stable.

In Fig. 4 we investigate these scenarios and their real-time ramifications in detail. We compute the completely informed reproduction number

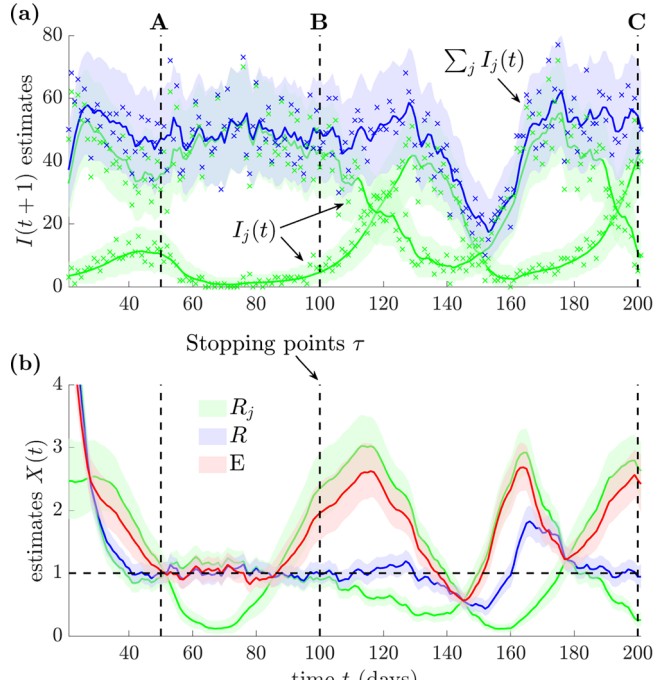

**Fig. 3 | Estimates of time-varying reproduction numbers with diverse R=1 scenarios.** We use renewal models (see 'Methods') to simulate epidemics in two regions ($p = 2$) with Ebola virus dynamics under generation time distributions from ref. 46. We plot (**a**) the time-series of the inferred one-step-ahead incidence $I(t + 1)$ for each region (green) and in total (blue), with actual incidence data shown as crosses. A flat one-step-ahead estimate means that the epidemic appears (critically) stable at that time. We identify three points A–C of apparent overall stability (black, dashed). We present time-varying reproduction number estimates (**b**) for the groups $R_j$ (green), the overall epidemic (based on the total incidence, blue) and estimates of the risk averse $E$ (red). We use $X(t)$ as broad notation for reproduction number estimates. In the flat total incidence regions of A–C, we have $R \approx 1$. We investigate each of the three regions of stability as key scenarios in this work. We derive all estimates from the EpiFilter package[28] with shaded areas representing 95% credible intervals in (**a**, **b**).

posterior estimate at our scenario timepoints $\tau$ and the expected probability of growth as new data emerge, consistent with practice during unfolding outbreaks. The posterior distribution given all data up to time $\tau$ is $\mathrm{P}\left(X|I_1^\tau\right)$, where $I_1^\tau$ is a vector of all incidence data from times 1 to $\tau$. All estimators process this data differently and are obtained by applying[28]. The posterior estimate of $R_j$ uses the component of $I(t) = I_1(t) + I_2(t)$ that informs on group $j$, $R$ uses $I(t)$ and $E$ uses both $R_j$ estimates. We plot data and posteriors respectively in panels (a, d, g) and (b, e h) of Fig. 4. We use the $\mathrm{P}\left(X|I_1^\tau\right)$ to infer the probability that total incidence $I(t)$ is growing $\mathrm{P}(I(\tau + h) > I(\tau))$ for a horizon $h$. These are panels (c, f, i) of Fig. 4 and are computed for $E$ and $R$ from $I(t)$ and for the groups by summing their local projections (the ground truth).

We find that scenario A is (roughly) genuinely stable. Both groups have their posterior mass near 1 and projections grow relatively slowly (we set a growth probability of 0.5 as a baseline for stability). In contrast, B-C present two key settings in which the $R$ threshold is misleading. B and C have $R$ around 1 but also one group with large $R_j > 1$ and a growth probability that is sizably higher than 0.5 and substantially above what $R$ indicates. The mechanisms underlying B-C are different. In B, the group with larger incidence is stable and this, together with infection stochasticity, masks the growth from the other group. Scenario C instead has groups with contrasting dynamics. The larger incidence group is declining and the smaller one is resurging such that their sum is stable. Unlike $R$, $E = 1$ accurately reflects stability amid the group uncertainties and heterogeneities in B-C but agrees with $R$ when all $R_j = 1$ as in A.

## Empirical data illustrate false stability thresholds

We have provided mathematical and simulation-based evidence that the popular thresholds of $R = 1$ and max $R_j = 1$ may often provide misleading assessments of epidemic stability. Here we demonstrate that these problems are apparent in real-world, empirical datasets and that $E = 1$ practically serves as an improved threshold for signifying overall or global stability. We consider a case study of the COVID-19 pandemic across 7 provinces of the Veneto region of Italy (data taken from ref. 17) and present our findings in Fig. 5. There we identify and analyse three key periods in which the estimated $R = 1$, around Aug 30th, Sep 15th and Nov 30th, 2020. We compute global statistics $X$ and the probability of $X > 1$ in alignment with previous analyses in this study and assess the resulting real-time indicators of stability.

We find that these three periods represent real-world examples of the scenarios discussed in Fig. 4. During Aug 30th we infer approximate $R = 1$ but a clear $E > 1$. From the group level incidence, we see that the province with the largest infections is declining but most of the provinces with small incidence are growing. This generalises scenario C of Fig. 4, where the dominant group obscures resurgent signals. $R$ and $E$ similarly disagree around Sep 15th, but now the provinces have comparable incidence levels and declining provinces counteract growing ones, mirroring scenario B from Fig. 4. In both periods max $R_j$ is above 1, overly sensitive to noise and unsuitable for guiding decisions. The Nov 30th period features genuine stability with $E = R = 1$ and even the max $R_j$ statistic overlaps 1. This reflects scenario A from Fig. 4. Figure 5 confirms that $E = 1$ can be a viable real-time epidemic stability threshold.

In Supplementary Note 1, we provide an additional empirical study of COVID-19 cases in the USA state of Texas. We consider incidence curves of the counties that compose that state (we take the top 24 counties by total cases) and estimate both $R$ and $E$. We again find clear periods in which the estimated $R = 1$ but several counties have growing incidence (and $E > 1$). We also identify periods where the estimated $E = 1$ but $R < 1$. Here the risk-averse reproduction number offers a better stability indicator as most counties are stable (with a few weakly growing). There is one county with a large, declining incidence, which is overly shaping the estimated $R$, forcing it confidently below 1. We do not show max $R_j$ as it does not help distinguish stability. Our supplemental results confirm that the phenomena we identify are not anomalies but rather a general risk when aggregating cases across heterogeneous regions.

## Epidemic thresholds outside of real-time, data-limited settings

Our analyses focused on real-time, rapid response applications, in which regions or locations are treated independently so that interactions are neither modelled nor informed by auxiliary data from contact matrices or mobility indices[3,4]. Neglecting these interactions is standard during unfolding outbreaks because these auxiliary data are frequently unreliable, unavailable or changing due to unexpected behaviours[20,21]. We therefore developed $E$ to refine best practices in these applications. Our approach also reflects federated surveillance and control settings[22], where because of differing health systems and policies (e.g. on data sharing, vaccine stockpiling), locations choose to act swiftly and independently of neighbours using easily interpreted and computed statistics. All of these properties are built into $E$.

In instances where explicit modelling of interactions is required, for example, if groups reflect age structure rather than geography, sophisticated methods have been developed to leverage contact and mobility data[8,15–19]. When such data are accurate, we do not expect $E$, which uses less information, to outperform these approaches. However, we may wonder whether $E$ can still perform well. In Supplementary Note 2, we develop mathematical arguments, extending the renewal models and transfer functions above to include interactions, and show that $E = 1$ is often a reasonable approximator of the epidemic threshold $T$ that has information on all the interactions among locations (parametrised in our analysis with migration probabilities).

We find that whenever migration is symmetric, local reproduction numbers converge or one location dominates migration (unidirectional flow), $E$ performs well when compared to $T$ (see Supplementary Note 1). In the last case, the local $R_j$ definition changes, reflecting the complexity of modelling interactions at multiple scales. $E$ may be less accurate when

**Fig. 4 | Growth projections from diverse epidemic scenarios with estimated $R \approx 1$.** We examine the scenarios A-C from Fig. 3. For each scenario, we plot incidence values on (**a, d, g**) (crosses represent data points and solid lines plot smoothed means) for each group (green) and their sum (blue). Total incidence is approximately stable for all scenarios, but the groups showcase diverse infection patterns. We compute posterior estimates given all past incidence data up to the endpoints $\tau$, denoted $I_1^\tau$ (**b, e, h**)) for the group reproduction numbers $R_j$ (green) and resulting global statistics $X$ of the overall (blue) $R$ and risk-averse (red) $E$ reproduction numbers. In all scenarios the posterior of $R$ is close to 1 (dashed black). This occludes important resurgences for groups in scenarios B and C. In (**c, f, i**), we explore ramifications of these posteriors by computing probabilities that the total incidence projected over a horizon $h$, $I(\tau + h)$ is larger than past incidence $I(\tau)$ (averaged over the last half-week from $\tau$ to reduce noise). Group-based projections (green), obtained by summing the $I_j(\tau + h)$ projected for every group from its $R_j$ posterior, define the expected ground truth dynamics. These dynamics are more reliably signalled at coarse scales by projections from $E$ (red) than from $R$ (blue) (these both use total incidence (blue) from the left panels). Estimates are from the EpiFilter package[28] and shaded areas are 95% credible intervals.

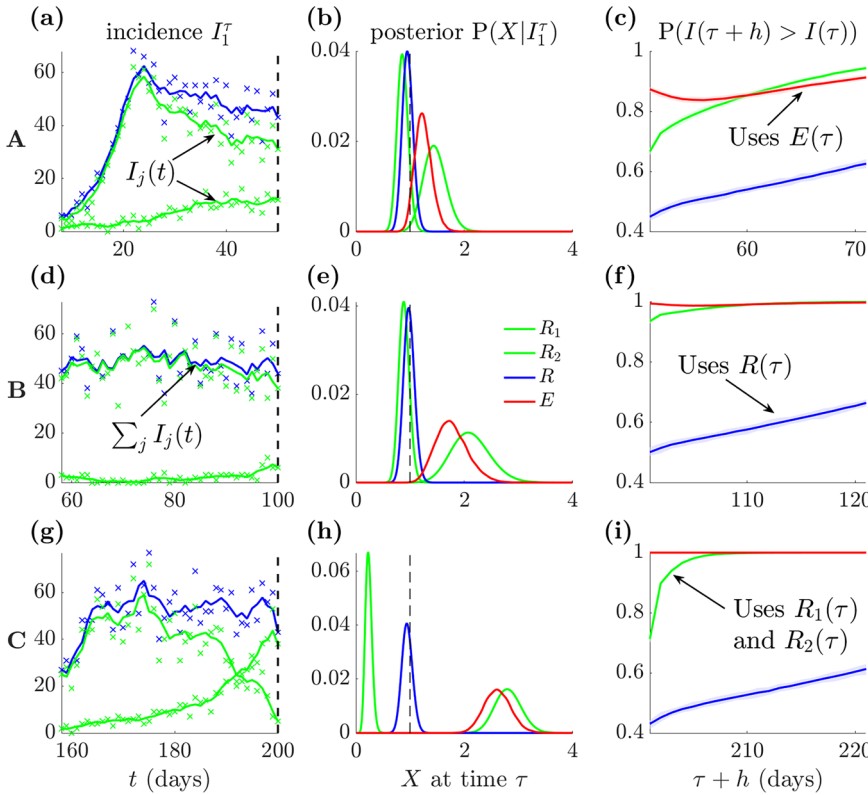

interactions are strong within a region and limited but still significant among regions. The standard $R$ is also a valid statistic across all these settings and is outperformed by $E$. This resilience in performance adds confidence in $E$ and we highlight that in our empirical studies $E$ improved on current best practice, even though some level of interactions will likely underlie these data (Veneto, Italy and Texas, USA). Ultimately, our study is about balancing descriptive power, assumptions and speed (e.g. ignoring interactions may be less harmful than getting them wrong). Striking the right balance, especially across scales, remains an open challenge in epidemic modelling.

## Discussion

Questions have been raised before about the accuracy and meaningfulness of reproduction numbers for tracking disease transmissibility[24,37–39]. Various works have appraised the formulation, estimation and adequacy of these statistics, especially under heterogeneous group dynamics and contact patterns[18,38,40] or when using data of varying availability and quality[31,37,41]. Despite these critiques, the interpretation that a reproduction number of 1 is the threshold of epidemic stability, whether derived from a ratio over infections ($R$ or similar)[42,43] or the maximum eigenvalue of a next generation matrix approach (related to max $R_j$)[8,15], has remained a fixture across epidemiology. Even studies exposing how stated values of reproduction numbers can be deceptive, strongly model-dependent and even meaningless in some instances[39,42], often agree on this threshold.

Here, we have revealed that common formulations of reproduction numbers, particularly in the context of real-time outbreak response where knowledge of mobility or contact rates among regions is unknown or hard to update, yield misleading stability thresholds. This extends the above critiques and explicitly considers when longstanding measures of epidemic stability can be untrue. We showed that even if total infections are roughly constant, $R$ in heterogeneous settings may cause biased and false projections of growth, misrepresenting overall epidemic risk. We exposed two consequential scenarios where $R = 1$ can conceal growth: if the resurgent groups (i) are masked by infections from stable groups or (ii) counteracted by groups with declining infections. We found that scenarios (i)–(ii) can easily

occur in practice (appearing in empirical data) and make $R$ a lagging indicator of exponential growth[9,10,17].

In contrast, we demonstrated that next generation matrix type statistics like max $R_j$, which consider group heterogeneities, can be highly sensitive to group stochasticity, leading to overly-conservative stability thresholds. Although an inferred max $R_j = 1$ guarantees overall stability (and is theoretically correct), achieving this is often impractical given the varying noise in infections, group estimates and uncertainty from data availability. Improved methods that combine next generation matrix theory with standard $R$ type averaging (as in ref. 8) do improve inference, but in those cases, the maximum eigenvalue type approach tends to somewhat agree with $R$ at the threshold of 1. Consequently, public health policy decisions informed by $R = 1$, max $R_j = 1$ and related variants may be at risk of being mistimed and inefficient[30].

By reframing both $R$ and max $R_j$ as extremes within a continuum of possible epidemic stability statistics, we uncovered a suitable compromise—the risk averse reproduction number $E$. This was introduced in ref. 12 to minimise the worst case uncertainty among group $R_j$ via experimental design but emerged here as an intermediate member of this continuum. Crucially, we found $E$ (and likely other members of the continuum) can better distinguish between genuinely stable scenarios (in which we recover $E = R = 1$) and scenarios where group or location heterogeneity only appears to lead to stability. The latter is obvious when $R = 1$ but $E > 1$ and max $R_j > 1$.

Consequently, experimentally designed statistics that leverage both the local (group level) and global (aggregate level) signals can generate more reliable thresholds of stability for informing outbreak response, exposing patterns obscured by $R$ but moderating dynamics amplified by max $R_j$. However, these benefits require identifying the heterogeneous groups (and stratifying the infections at the group level) in real time. While grouping by geographical or demographic features is possible, provided data are collected at those scales, it is difficult to disaggregate groupings by immunity levels, behaviour or other less conspicuous heterogeneities. In limited data settings or when grouping is unknown, both $E$ and next generation matrix type methods cannot be computed. Accordingly, $R$ then remains the best means of tracking transmissibility.

**Fig. 5 | False stability thresholds for COVID-19 in Italy.** We examine COVID-19 dynamics across $p = 7$ groups, which are the provinces of Veneto, Italy. Data are the incidence of new positive tests from ref. 17, which we split into two time-regions of interest (A-B) and use renewal models with generation times from ref. 47. **a, b** plot incidence $I_j(t)$ across the provinces (green) and the total incidence (black), normalised to the maximum $I_j(t)$. We estimate the next generation type max $R_j(t)$ (cyan), standard $R(t)$ (blue) and the risk averse $E(t)$ reproduction number (red) using EpiFilter (**c, d**) and compute the probability that these statistics (broadly denoted $X(t)$) are above 1 (**e, f**). A: we observe two periods with $R \approx 1$ and $E > 1$. In the first (about Aug 30th), most of the provinces show growing infections, but the one with the largest incidence declines. This mirrors scenario B from Fig. 4. The second period (about Sep 15th) is analogous to scenario C from Fig. 4, with both declining and growing infections in provinces summing to a near constant incidence. The max $R_j$ statistic is too noise sensitive and always signals epidemic growth. Across both periods, $E$ correctly infers the stability trends, balancing the misleading estimates from $R$ and max $R_j$. B: a period of genuine stability. The provinces all have relatively stable incidence (around Nov 30th) and all global statistics largely agree (though max $R_j$ is still too sensitive). This occurs because epidemics are synchronised and is akin to scenario A from Fig. 4.

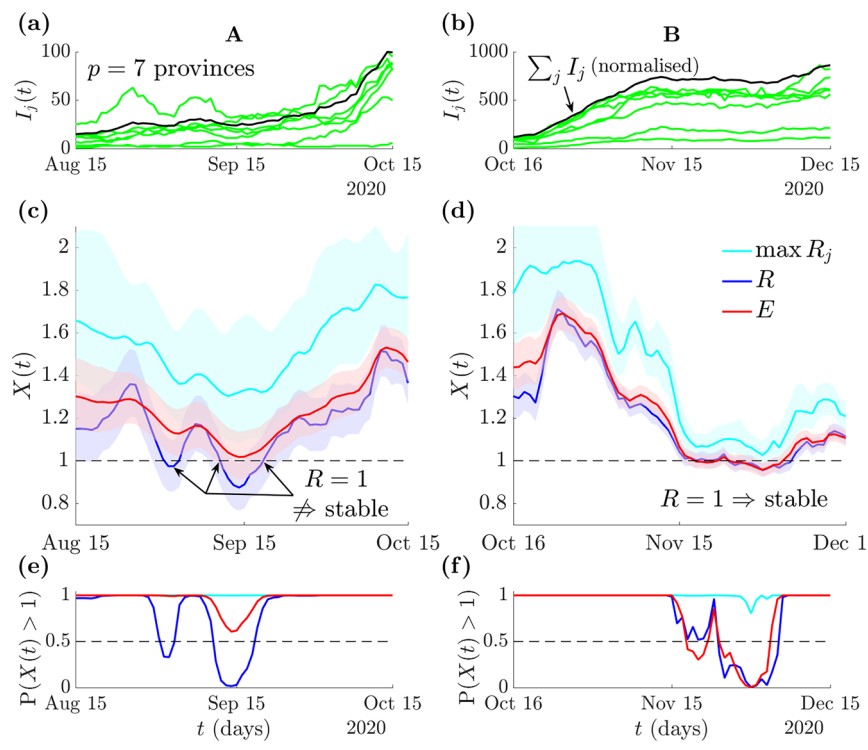

As we need group-level information to infer $E$, one may question the added value of global statistics for decision-making relative to the already computed $R_j$. If groups are homogeneous, then $R_j = 1$ correctly indicates stability. However, issues arise when assessing wider patterns that effect interventions. For example, if $p = 5$, how many groups should have $R_j > 1$ before we can reliably justify a widescale action (e.g. lockdowns)? This information is also valuable for already stable groups as failing to act globally given the likely connectivity and mobility among individuals can be problematic for curbing spread. Global knowledge also helps shape local strategies such as when the stable groups or locations need to enforce travel restrictions[44].

Stability thresholds are fundamental benchmarks for implementing or releasing interventions and pinpointing epidemiologically important shifts. As these shifts frequently emerge at group levels, we recommend that future response efforts should prioritise (a) robustly detecting the heterogeneous subgroups within a population (b) estimating transmissibility within those well-mixed groups (to obtain $R_j$) and (c) merging those estimates into balanced global statistics like $E$. An $E = 1$ more reliably indicates overall epidemic stability and practically achieves what $R = 1$, commonly and overconfidently, only estimates theoretically.

## Methods
### Renewal models and averaging inherent in R numbers
The effective reproduction number $R$ of an infectious disease is commonly inferred over large scales, for example across a country or nation, by modelling the epidemic at this scale as a renewal process that ignores all heterogeneity[1]. This model assumes that the incidence $I$ is the noisy product of $R$ and the total infectious force of the disease $\Lambda$. For Poisson (Pois) noise this model is $I \sim \text{Pois}(R\Lambda)$ Here, $\Lambda$ is the convolution of past incidence with the generation time distribution i.e. $\Lambda = \sum_{x=1}^{t-1} G(x)I(t - x)$, with $G(x)$ as the probability that an infection is transmitted in $x$ time units, and defines how many infectious individuals are able to generate secondary infections in the population. The predominant method for inferring $R$[3] results in a gamma (Gam) posterior estimate distribution for $R$ (ignoring prior distribution terms) of Gam $(I, \Lambda)$. The posterior mean estimate of $R$ is then $I\Lambda^{-1}$ and its variance is $I\Lambda^{-2}$[3].

A simple extension of this renewal model has also been applied to describe the epidemic at smaller scales, such as when we consider local epidemics in regions that compose our study country[1]. Assuming $p$ groups or regions, this approach describes the incidence of infections independently in each group $j$ as $I_j \sim \text{Pois}(R_j\Lambda_j)$ with $\Lambda_j = \sum_{x=1}^{t-1} G_j(x)I_j(t - x)$. The $G_j(x)$ compose the generation time distribution of group $j$. The effective reproduction number of this group has posterior estimate distribution Gam $(I_j, \Lambda_j)$. The mean estimate of $R_j$ is $I_j\Lambda_j^{-1}$[3]. This extension is commonly used to provide localised reproduction number estimates[27,29]. However, it is uncommon to consider what these two descriptions of the epidemic imply.

Aggregating the incidence from all $p$ groups yields $I = \sum_{j=1}^{p} I_j \sim \text{Pois}(\sum_{j=1}^{p} R_j\Lambda_j)$. Because this method often assumes a fixed generation time distribution among groups i.e. all $G_j(x) = G(x)$[27,29], it follows that $\Lambda = \sum_{j=1}^{p} \Lambda_j$. Equating this with the earlier expression for $I$ we see that $R$ is a weighted mean of the $R_j$ as in Eq. (1), with weights $\Lambda_j(\sum_{i=1}^{p} \Lambda_i)^{-1}$. Consequently, irrespective of the inference method applied, standard $R$ estimates will possess this weighted-average relationship. The popular approach of[3] mentioned above also reveals the inherent averaging within $R$ because if $R_j \sim \text{Gam}(I_j, \Lambda_j)$ then $R = \sum_{j=1}^{p} \Lambda_j(\sum_{i=1}^{p} \Lambda_i)^{-1} R_j \sim \text{Gam}(I, \Lambda)$ This representation of $R$ underlies Eq. (1) and subsequent simulations in Figs. 1–4.

### Epidemic transfer functions and the max Rj statistic
Control theory is a field concerned with regulating the stability and performance of dynamical systems. The renewal processes defined above are examples of linear systems and their local and global stability properties are completely measured by the locations of their poles in the complex $s$ plane. Poles are solutions to the denominator of the transfer function of the system[33]. We construct this following[34]. If a system has at least one pole with a positive real part then it is unstable (i.e. infections grow unboundedly). Critical stability i.e. the situation that is conventionally described by $R = 1$

and $r = 0$, occurs only when the real parts of all poles are non-positive with at least one pole having zero real part.

Consider a group with an initial imported infection represented by the delta function $\delta(t)$ and basic reproduction number $R_{0j}$ (or equivalently some constant approximation to the effective reproduction number). The deterministic, continuous time version of the renewal process that describes the infections in that group is $I_j(t) = \delta(t) + R_{0j} \int_0^t G_j(t-x) I_j(x)\, dx$. We take the Laplace transform of this equation. For a signal $v(t)$ in the time domain, this is $\int_0^t v(x) e^{-sx}\, dx$ in the $s$ domain. We rearrange to get $I_j(s) M_j(s)^{-1} = \left(1 - R_{0j} G_j(s)\right)^{-1}$[34]. The argument $s$ denotes that our signals are now in the complex frequency domain and $M_j(s)$ indicates that any import time series is usable. This ratio of output infection to input is the transfer function.

We consider a delta function (initial seed infection), which has Laplace transform $M_j(s) = 1$. Accordingly, the transfer function also directly defines the infections in the group. Poles, which are the same regardless of the chosen $M_j(s)$, are solutions to $R_{0j} G_j(s) = 1$. The dominant pole of this system is always real, has the largest real part and equals the epidemic growth rate[32,34]. Hence, if this pole or the growth rate is 0, we have critical stability. This can be shown to require that $R_{0j} = 1$[25]. The other poles shape other characteristics of the infection trajectory (e.g. peaks and oscillations in response to interventions or other imports) and this framework completely describes the renewal epidemic dynamics, assuming that the depletion of susceptibles is sufficiently slow that its effect is absorbed within $R_{0j}$.

The benefit of this framework is for pinpointing the conditions for critical stability when we want to consider the global dynamics of all $p$ groups. We assume, as in many real-world and real-time situations, that we do not know the connectivity among groups and so can easily define $I(s) = \sum_{j=1}^p I_j(s)$ as the transfer function of the entire heterogeneous system. If we did know the connectivity, this transfer function would be modified to include group interaction terms[45]. However, even under this generalisation the poles still completely describe the dynamics of the system and our approach is broadly applicable. This framework leads to Eq. (2) and because we need all our poles to have non-positive real parts for critical stability, we find the maximum group growth rate must be 0 and hence max $R_j = 1$ for critical stability.

This approach while grounded in control systems theory overlaps the next generation matrix and related methods such as[8,23], which focus on eigenvalues instead of poles. In many instances these are exactly the same, but the transfer function also pays attention to the other poles and the Laplace transforms are valuable for describing all the system dynamics (e.g. not only do we examine the other poles but can assess how they influence e.g. the size of the endemic infection load when the system is critically stable). Another benefit is that since we clarify how local poles contribute to global ones, we know conditions for stability for various architectures of groups (e.g. series or parallel connections among groups) immediately and can even assess how control applied to imported and local cases shape stability[45].

### General global statistics X and risk averse E

There are several ways of examining the stability properties of epidemics that spread across heterogeneous groups. The above subsections considered two of the most popular: $R$ (or E[$R$] when infection stochasticity is included with E denoting expectation) and max $R_j$ (or max E[$R_j$]). The first is used directly for real-time analyses at broad spatial and socio-demographic scales and is the predominant time-varying measure of stability in epidemiology. The latter, often in a related next generation matrix form, is more frequently used for basic reproduction numbers and when connectivity is known but is also an important measure of heterogeneous stability. As we applied both under the same assumptions, we aimed to extract the key operation that each measure might be applying in relation to the group level $R_j$.

Our suite of global statistics, $X$, is obtained by recognising that most statistics are essentially a type of weighted average over the $R_j$. The central distinction is how weights are chosen, with $R$ performing averaging based on the relative total infectiousness of groups and max $R_j$ using a limiting weighting that is representable as a power mean with a power that tends to infinity. This insight led to the generic formula of $X$ in Eq. (3). The risk averse reproduction number $E$ was originally obtained in ref. 12 by solving the E-optimal experimental design of minimising the maximum uncertainty among $R_j$ estimates. This involves constructing the Fisher information matrix for the overall (global) renewal model, which has $p$ eigenvalues of form $\alpha_j R_j^{-1}$. $E$ results by maximising the smallest $\alpha_j R_j^{-1}$ subject to a constraint of $C = \sum_{j=1}^p \alpha_j$. This yields a statistic of $\sum_{j=1}^p \frac{\alpha_j}{C} R_j$ with the optimised $\alpha_j = \Lambda R_j \left(\sum_{i=1}^p R_i\right)^{-1} = \Lambda \omega_j(1)$. Combining these gives Eq. (4), which emphasises resurgent group dynamics but acknowledges that resurgence at small infection counts (where less data are available) leads to larger estimate uncertainty[12].

## Data availability

All data used are publicly available and also accessible at https://github.com/kpzoo/stabilityR. All other relevant data are available from the authors upon request.

## Code availability

We provide MATLAB code at https://github.com/kpzoo/stabilityR to reconstruct the figures and reproduce all the analyses. The release is https://github.com/kpzoo/stabilityR/tree/v1.0.

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

## Acknowledgements

K.V.P. and A.C. acknowledge funding from the MRC Centre for Global Infectious Disease Analysis (reference MR/X020258/1), funded by the UK Medical Research Council. This UK-funded grant is carried out in the frame of the Global Health EDCTP3 Joint Undertaking. UO was supported by a grant from Tel Aviv University Center for AI and Data Science 417 (TAD) in collaboration with Google, as part of the initiative of AI and DS for social good. The funders had no role in study design, data collection and analysis, decision to publish, or manuscript preparation.

## Author contributions

Conceptualisation, investigation, formal analysis, writing (original draft preparation), funding acquisition, supervision: K.V.P. Software, visualisation, project administration: K.V.P. and U.O. Methodology, validation, writing (review and editing): K.V.P., M.S., A.C. and U.O.

## Competing interests

The authors declare no competing interests.
