## [Transparent Peer Review file · Communications Physics]

The $R = 1$ threshold can misclassify epidemic stability

Corresponding Author: Dr Kris Parag

Version 0:

Reviewer comments:

Reviewer #1

(Remarks to the Author)

In this paper the authors demonstrate the limitations of the classical definition of the reproduction number as a criterion for epidemic behaviour, primarily the boundary condition $R=1$ signifying epidemic stability. Instead they propose the risk-averse reproduction number E , which weights the contributions of the group-specific reproduction numbers as the better criterion compared to the classical R , but which requires more detailed information about the epidemic transmission and behaviour in a heterogeneous population compared to the standard approach.

The authors do a great job at summarising both the theoretical aspects underpinning the derivation and subsequent comparison of the three metrics R , E and $\max(R_j)$. The figures demonstrate visually the existence of some scenarios in which the metrics R and $\max(R_j)$ fail to correctly determine epidemic behaviour (Fig 5), as well as the performance of the three metric in comparison to one another (Fig 4).

My only comment would be relating to the computation of R and R_j . As the authors pointed out, they do not consider any interactions between groups, to the point where in the computation of R_j , only the daily incidence data in group j is used in the computation of Λ_j (it was not very clear, please add the formula for improved clarity). However, this is quite a strong assumption for the vast majority of scenarios; indeed, if we were to use this proposed approach for an age-stratified population rather than region-structured, modelling the interactions between these age groups is not only desired, but rather expected. In light of other studies that look into heterogenous population renewal equations [1,2], which define R and R_j in terms of the contact matrix, I was wondering if the equations 1 - 4 cannot be updated to better account for that? From a preprint of mine that will be sent to publication soon and extends [2], I got the equation in the attached supplementary file.

And if that change can be easily implemented as I think it can, how does that impact the suitability of E as an alternative metric to R ?

Minor Observation, on page 5, there should not be a comma in : "these false indicators of stability, may". Additionally, on page 6 can you better explain what you mean why "If we assume a seed infection starts the epidemic in all groups, then $M_j(s) = 1$."?

Overall it was a wonderful read and the results are of great interest to the infectious disease modelling community at large.

[1]Boldin B, Diekmann O, Metz JAJ. 2023 Population growth in discrete time: a renewal equation oriented survey. J. Differ. Equ. Appl. 1–29. (doi:10.1080/10236198.2023.2265499)

[2]Bouros Ioana, Thompson Robin N., Gavaghan David J. and Lambert Ben 2025The time-dependent reproduction number for epidemics in heterogeneous populationsJ. R. Soc. Interface.2220250095 (doi:10.1098/rsif.2025.0095)

Reviewer #2

(Remarks to the Author)

See attached PDF

Reviewer #3

(Remarks to the Author)

The authors investigate cases where the epidemic stability threshold

R

=

1

$R=1$ may lose its conventional meaning. This topic is interesting and potentially valuable. However, the current manuscript suffers from structural and clarity issues that significantly hinder comprehension. Therefore, I do not recommend publication in its current form. A thorough revision is needed before the manuscript can be reconsidered. My specific comments are as follows:

Manuscript Structure:

The overall structure should be reorganized for clarity. For example, in Fig. 1, the compartment E appears, yet its meaning is not introduced until Eq. (4) later in the text. Such delayed definitions make it difficult for the reader to follow the model formulation. I suggest introducing all variables and compartments before presenting the figures and model equations.

Conceptual Clarity in Key Sections:

The arguments in the sections "Reproduction numbers may provide false positive indicators of stability" and "Next-generation matrices may provide false negative indicators of stability" are unclear. These are central claims, yet the manuscript lacks detailed explanation and rigorous justification. Please provide clearer theoretical arguments, illustrative examples where necessary, and a step-by-step explanation to support these claims.

Formatting and Presentation:

There are several formatting issues that should be corrected. For instance, all equation numbers should be right-aligned for consistency with standard mathematical formatting. A careful proofreading of the entire manuscript is recommended.

Version 1:

Reviewer comments:

Reviewer #1

(Remarks to the Author)

I approve all changes made to the manuscript by the authors in response to the reviewer comments. I have seen one small typo: on page 2, "as both a prospective or retrospective", replace or with and.

Reviewer #2

(Remarks to the Author)

I believe that the revised version successfully addresses the reviewers' comments and is acceptable for publication.

I only have two remarks:

- In the legend of Fig. 3 the authors state "We identify three regions of apparent overall stability (black, dashed)". I think that the word "region" is confusing, both because "region" is used in the previous sentence to denote a spatial unit, and (more important) "region" generally denotes an extended time length (that one does not see in the Figure), while here I would say "three times of apparent..." or "three time points of apparent..."
- In the new paragraph (which I appreciated) "Epidemic thresholds outside of real-time, data-limited settings", the authors state "whenever migration is symmetric... E performs well when compared to T". However, looking in the Supplement at case 2, one reads " $T=1$ requires $(\delta+\epsilon)(1-\rho)\approx 0$ while $E=1$ requires $0.5(\delta+\epsilon)\approx 0$ ". If ρ is close to 1 (as I would expect: most contacts in one's own group), say $\rho=0.9$, the threshold for E is off almost by a factor 2. I would not say it performs well in such a case. Also it seems that the use of E works best when $\rho=1/2$, meaning that essentially one has a well-mixed population with two types differing in transmissibility, but not very well when transmission between groups is small. I believe that this observation deserves a comment.

Reviewer #3

(Remarks to the Author)

The authors have addressed all my questions. I suggest it for publications.

Responses to Reviewers: COMMSPHYS-25-1582

Editor:

We hope you will find the referees' comments useful as you decide how to proceed. Should further stochastic simulations and analysis allow you to address these criticisms, we would be happy to look at a substantially revised manuscript.

We thank the reviewers and the editor for constructive and insightful comments. We list our responses to each comment and corresponding changes to the main text below. This includes details on further simulations and analyses. We provide an additional version of the main text with changes tracked to highlight our revisions. All references made below are in a separate bibliography at the end of this document.

Reviewer #1:

The authors do a great job at summarising both the theoretical aspects underpinning the derivation and subsequent comparison of the three metrics R , E and $\max(R_j)$. The figures demonstrate visually the existence of some scenarios in which the metrics R and $\max(R_j)$ fail to correctly determine epidemic behaviour (Fig 5), as well as the performance of the three metrics in comparison to one another (Fig 4).

Thanks for the positive comments and the suggestions, which have strengthened the paper.

1) My only comment would be relating to the computation of R and R_j . As the authors pointed out, they do not consider any interactions between groups, to the point where in the computation of R_j , only the daily incidence data in group j is used in the computation of Λ_j (it was not very clear, please add the formula for improved clarity).

We have added the explicit formula for Λ_j in the Methods, where this term is defined. In the first Results section, where Λ_j is introduced, we have clarified that it depends on past I_j only.

2) However, this is quite a strong assumption for the vast majority of scenarios; indeed, if we were to use this proposed approach for an age-stratified population rather than region-structured, modelling the interactions between these age groups is not only desired, but rather expected.

We agree that neglecting interactions between groups may be a strong assumption in certain scenarios. However, we focus on geographically defined "demes", where interactions should be weaker than among age groups and specifically on tracking transmission during unfolding outbreaks in real time, where this assumption is not only standard but often predominant and necessary because interactions among demes are unknown, misspecified, unmeasurable or change unexpectedly (e.g. due to poorly understood population behaviours). During COVID-19, which featured the highest resolution infectious disease data in history, most estimates of transmissibility assumed non-interacting demes (e.g. highly popular software like EpiEstim and EpiNow2, which informed government policies) for several reasons:

- a. Lack of available, updating, contextual or accurate mobility, migration and interaction data. It is rare to have contact matrices and mobility indices that are representative of the region of interest and that update as the epidemic progresses. Many countries (e.g., LMICs) lack the infrastructure for such surveillance and even within a country data quality can vary substantially among local authorities. Transplanting contact matrices from other locations may introduce substantial biases as highlighted in [1,2]. Further, having access to detailed information (e.g., contact networks) may not improve $R(t)$ estimates as found in [3] and even when useful, mobility indices that are predictive pre-intervention can lose that power post-intervention due to behaviour or rebound effects, as explored in [4]. Neglecting spatial interactions can therefore sometimes be the lesser of evils.
- b. Federated surveillance and control. Frequently, due to differing health and surveillance systems, standards and policies (e.g. on data sharing, vaccine stockpiling) [5], locations or demes may choose to act independently of neighbours when targeting interventions. This was part of the motivation for non-interacting and popular $R(t)$ hotspot analyses in [6] and reflects that often a location prefers to base decisions on data mostly arising from that location. Interactions are only considered in the sense of imported infections. Control of imports entails travel restrictions while control of local infections involves contact-tracing or vaccination, for example. This type of breakdown is also handled in our framework by adjusting the R_j for imports (when this data are available) as in [7].
- c. Computational and mathematical complexity, especially when informing policymakers or the public. The joint modelling of locations and their contact frequencies (if known) is a demanding task for real-time applications [6], where information is needed daily and many downstream analyses may follow on from $R(t)$ estimates (e.g., counterfactual projections). Moreover, it is harder to interrogate such complex models and understand their estimate uncertainties. Importantly, when modelling contact structures (as we investigate below in our response to 3)) the idea of a local reproduction number is not easily defined and may change based on the magnitude of interactions.

Our goal is to provide real-time improvements to the methods usually applied because of a-c above, while maintaining interpretability, speed and minimising data needs. We also bridge the often-overlooked link between global and local estimates of transmissibility, when deciding between independently enacted, targeted actions and the need for a nationwide, synchronous action such as a lockdown. We have now summarised these points in the Introduction and in a new Results section to better clarify the motivation, focus, use-case and limitations of our approach. We also mention methods that integrate and improve $R(t)$ with contact and mobility data such as [8–11] and their value in settings where interactions are important. Hopefully our rationale for neglecting spatial interactions is clearer.

3) In light of other studies that look into heterogenous population renewal equations [1,2], which define R and R_j in terms of the contact matrix, I was wondering if the equations 1-4 cannot be updated to better account for that? From a preprint of mine that will be sent to publication soon and extends [2], I got the equation in the attached supplementary file. And if that change can be easily implemented as I think it can, how does that impact the suitability of E as an alternative metric to R ?

We copy the equation recommended by the reviewer below. While this is very interesting and appears to nicely convert a global $R(t)$ into a group-specific one, it cannot be easily integrated within our framework, which was designed to be agnostic to contact structure.

$$R_t^{(k)} = R_t \frac{\left(\sum_{i=1}^N \Lambda_t^{(i)}\right) \left(\sum_{j=1}^N C_t^{(jk)}\right)}{\sum_{i=1}^N \sum_{j=1}^N C_t^{(ji)} \Lambda_t^{(i)}}.$$

Including contact matrices in this way could form an interesting future study but is not within the scope of our current approach. Importantly, in our empirical examples we do not possess the $C_t^{(jk)}$ and we explored how simple changes from R to E , which use available data, can already markedly improve the decision-making information in real-time estimates. We also note there is no consensus yet on what works best when describing spatial interactions and the topic is an active area of research with emerging methodology as in [8–11].

That said, we do agree that understanding interactions is important and duplicate the analysis below in a new Supplement section. These validate a-c above about why accurate knowledge of migrations is valuable but inaccurate knowledge can be problematic. We also make clear in the Results section mentioned in 2) that we do not expect E to outperform directly derived $R(t)$ metrics that consider full knowledge of the migration or contact probabilities. However, we do improve estimates when such knowledge is unavailable or unreliable. We better qualify E (making clear its limitations and use when locations interact) and motivate our results better as suitable for low information, real-time policy-driven studies. The following text is now in the Supplement and we summarise key points resulting from it in response 5) to Reviewer 2.

We consider a heterogeneous renewal model broadly defined below with p_{kj} as the probability of an infection generated in location k migrating into location j .

$$I_j(t) = m_j(t) + p_{jj}R_j\Lambda_j + \sum_{k \neq j} p_{kj}R_k\Lambda_k. \quad (i)$$

Here $\Lambda_j = \int_0^t G(t-x)I_j(x) dx$ and the $m_j(t) = 0$ except in the location where the epidemic starts, then $m_j(t) = \delta(t)$. We assume the same generation time distribution across locations for notational convenience and use deterministic equation forms (though including Poisson noise is straightforward). As the migration probabilities are conserved $\sum_j p_{kj} = 1$.

An important point is immediate from (i). If we look at the total infections $\sum_j I_j(t) = I(t)$ and collect our sums of migration terms, then we obtain $I(t) = \delta(t) + \sum_j R_j\Lambda_j$. Comparing to the commonly assumed global model $I(t) = \delta(t) + \Lambda R$, we find that under any migration scheme the standard R is unchanged from the formulation that neglects interactions. Consequently, in these cases this popular R may still be interpreted as a meaningful baseline.

We can take Laplace transforms of (i) to get (ii) below. This yields a matrix of transfer functions.

$$I_j(s) = m_j(s) + p_{jj}R_jG(s)I_j(s) + \sum_{k \neq j} p_{kj}R_kG(s)I_k(s). \quad (ii)$$

To better explore the diversity of transmission possibilities emerging from interaction, we solve the equations for 2 demes or locations only. However, similar analyses can be applied to larger

group numbers. We denote p_{11} as p_1 and note that $p_{12} = 1 - p_1$. Symmetrical arguments apply for p_{22} and p_{21} leading to the characteristic polynomial $\Delta(s)$ for this system in (iii).

$$\Delta(s) = (1 - p_1 R_1 G(s))(1 - p_2 R_2 G(s)) - (1 - p_1)(1 - p_2) R_1 R_2 G^2(s). \quad (\text{iii})$$

All the poles of the heterogeneous renewal model of (i) are the solutions of $\Delta(s) = 0$ and the dominant pole is the overall growth rate. For a critical dominant pole (zero growth), we require the parameters in (iii) to satisfy $\Delta(s=0) = 0$. This yields the condition (iv) i.e., if our p_j and R_j satisfy this relation then we are certain that the true epidemic threshold T is at 1.

$$p_1 R_1 + p_2 R_2 - 1 = (p_1 + p_2 - 1) R_1 R_2. \quad (\text{iv})$$

As (iv) uses all possible information, we do not expect E to perform as well as the resulting metric obtained from this condition. However, we can show the value of E on key conditions below. We define the threshold of stability by rearranging (iv) into a form $T = 1$.

1. Antisymmetric migration: $p_1 = 1 - p_2$. In this case, the threshold stability satisfies $p_1 R_1 + p_2 R_2 = 1$, which is a weighted mean of the local reproduction numbers. Under any choice of p_1 we recover a version of Fig 1 from the main text, where E is better than R . However, the fully informed $T = p_1 R_1 + p_2 R_2$ will always perform best but requires us knowing the migration probabilities as well as local reproduction numbers.
2. Symmetric migration: $p_1 = p_2 = \rho$. Substituting this into (iv) we get $\rho(R_1 + R_2) - 1 = (2\rho - 1)R_1 R_2$. If we take small deviations (δ, ϵ) about 1, so that $R_1 \approx 1 + \delta$ and $R_2 \approx 1 + \epsilon$, then $T = 1$ requires $(\delta + \epsilon)(1 - \rho) \approx 0$ while $E = 1$ requires $0.5(\delta + \epsilon) \approx 0$. Hence, when $\rho \neq 1$, E locally approximates the correct threshold condition. If $\rho = 0.5$, the approximation becomes exact. If instead $\rho = 1$, we recover the non-interacting case.
3. Dominant (unidirectional) migration: $p_2 = 1$. Critical stability requires $p_1 R_1 + R_2 - 1 = 1 + p_1 R_1 R_2$. This is the same as the non-interacting case of the main text, but the local reproduction numbers are now $p_1 R_1$ and R_2 . If $p_1 = 1$, they are $p_2 R_2$ and R_1 . Adjusting the definition of E to these new local reproduction numbers recovers our results from the main text, but it is clear even unidirectional interactions change our notion of local spread.
4. Symmetric transmissibility: $R_1 = R_2 = \mu$. Now (iv) becomes $(p_1 + p_2 - 1)\mu^2 - (p_1 + p_2)\mu + 1 = 0$. This quadratic is solved by $T = \mu = 1$. Computing $E = \sum_j R_j^2 / \sum_j R_j = \mu$ we find E is precisely the true stability threshold for any migration probability.

Consequently, $E=1$ can in several instances be a good approximator for overall stability. While less accurate than T , E requires far less information, which makes it apt for real-time analyses. T and mobility or contact-matrix based methods are advantageous for retrospective analyses from high resolution data. However, if the p_{kj} are not well known or change due to unexpected dynamics, computing T is difficult and biases can emerge (e.g., 1-4 are notably diverse).

Minor Observation, on page 5, there should not be a comma in: "these false indicators of stability, may". Additionally, on page 6 can you better explain what you mean why "If we assume a seed infection starts the epidemic in all groups, then $M_j(s) = 1$."?

Thanks, we have fixed these issues and expanded that when a seed infection occurs, we can describe this input with a δ function. The Laplace transform of this δ is 1, yielding $M_j(s)$.

Overall, it was a wonderful read and the results are of great interest to the infectious disease modelling community at large.

Thank you!

[1] Boldin B, Diekmann O, Metz JAJ. 2023 Population growth in discrete time: a renewal equation oriented survey. *J. Differ. Equ. Appl.* 1–29. (doi:10.1080/10236198.2023.2265499)

[2] Bouros Ioana, Thompson Robin N., Gavaghan David J. and Lambert Ben 2025 The time-dependent reproduction number for epidemics in heterogeneous populations. *J. R. Soc. Interface.* 2220250095 (doi:10.1098/rsif.2025.0095)

Reviewer #2:

The main message of the manuscript under review is that, when a population is split into different groups, and separate incidence data are available for these groups, a summary statistics, named E , that averages in a suitable way the effective reproduction numbers R_j computed for each group, can be an appropriate indicator for guiding public health policy in real time, much better than the effective reproduction number R computed for the whole population. In order to support the use of E , the authors consider the example of the data of Covid-19 in the autumn 2020 in the Veneto region of Italy, as well as some simulations that appear to have been structured to yield incidence data somewhat reminiscent of those of the Veneto region. I think this makes an interesting contribution to the field. However, I also think that the manuscript needs be improved before publication.

Thanks for the positive appraisal and critique, which have notably improved the paper. We highlight, in case it was missed in the Supplement, that we analysed additional empirical data from 24 counties in the state of Texas, USA and found further support for using E .

1) A first observation, somewhat marginal, is that the authors introduce the formalism of transfer functions, in the Results (6th page of the text) and later in the Methods, without any obvious connection to the rest of the manuscript, except to state the well-known result that in theory the stability region requires $\max R_i < 1$.

Our rationale for using transfer functions was that it provides an established and interpretable link between reproduction numbers and poles. This allowed us to combine epidemic systems together easily (e.g., we can use this approach with other architectures such as regions that trade infections among one another) and clarify how overall system poles depend on the poles of constituent regions and hence how the overall growth rate relates to rates of growth in all the locations. While the overall result may be unsurprising, the clarity is valuable as our aim is not only to understand global dynamics but how local dynamics drive them. We have added text better explaining the rationale in that Results section and in the Methods.

Additionally, in our new Supplement section, investigating when locations are not isolated but do interact, we again make use of the transfer function approach to extract central insights about how local and global growth rates interact. This allows us to identify the true epidemic threshold (via the characteristic polynomial and its roots, which are the transfer function poles) under interactions and find cases when E is a good approximator and when it may be less suitable. See response 3) to Reviewer 1 above for a complete description of these new results, which also better connect to the transfer function development in the main text.

2) The main problem in the manuscript lies, in my opinion, in the lack of details about the simulations performed. Furthermore, I would have liked to see the effectiveness of E in model-based simulations where the ground truth is known and clear. From the text one can only guess how the simulations shown in Fig. 3 have been obtained. It is true that the code is available in Github, so one could read the code to understand it, but I would expect to get the main elements of the simulations from the text. I gather that the authors have used (slowly) time-varying reproduction numbers, varying asynchronously in the two groups, and that incidence data have been simulated drawing random variables (Poisson?) with mean given by the renewal formula. Possibly the assumed values for the reproduction numbers are close to the estimates, but definitely we are missing most details of the simulations.

We apologise that the simulations have not been described in sufficient detail in the text. We have now included more extensive and transparent information about the simulations in the Results sections. To summarise: the ground truth in Fig 3 are the local reproduction numbers R_j , which we now detail in the text. In one location it varies sinusoidally, while the other location uses a feedback algorithm to adapt transmissibility to uncover periods with stable incidence (this algorithm is now outlined in the text). The green credible intervals in the lower panel of Fig 3 provide Bayesian estimates of these R_j , which are known to minimise mean squared error under the established method we used [12]. These indeed estimate the ground truth well (the good fit of the one-step-ahead incidence also confirms this performance).

However, at the global level there is no objective ground truth as R , $\max R_j$ and E are all different ways of summarising the population-level infectiousness. The first is correct under a global renewal model assumption, the second emerges from next generation matrix or transfer function analysis and makes assumptions about transmissibility fluctuations. The last is what we propose as a meaningful compromise between those sets of assumptions. Local incidence counts are drawn from the Poisson renewal models (defined in the Methods) as the reviewer concludes. These counts are plotted in the upper panel of Fig 3. Fig 4 is a closer look at local and global estimates from key slices of the simulation in Fig 3. This simulation mimics how data emerges (stochastic infection counts) and how they are commonly analysed (estimates from renewal models at local and global levels).

3) Similarly for Fig.2 they state that they sampled 1,000 weighing from the Dirichlet distribution (presumably the mean a symmetric Dirichlet distribution, but do not specify its parameter); I understand that in this way weights w_i (between 0 and 1 that sum up to 1) are obtained. However, the authors need R_i whose average is (approximately?) 1; perhaps, they obtain $E[R_i] = pw_i$ and then simulate 5,000 samples of R_i (with an unknown distribution), but this is only my guess. The authors should provide some details.

We have now improved clarity by explaining all the steps in this Results section that are used to obtain the symmetric Dirichlet samples and eventual mean squared errors. We include the parameters, means of local R_j and explain how the Dirichlet settings allow us to control the heterogeneity among the reproduction numbers. The control of the mean of the R_j comes from the weights which set the I_j . The samples of the estimates of R_j use the I_j and Λ_j and follow from standard theory (the seminal EpiEstim paper). We set the total level incidence and total infectious force to be the same ensuring a global $R=1$ underlies every Dirichlet sample set.

4) Concerning Figure 2, I must add that MSE of the $R_j = \sum R_j^2 - 2\sum R_j + p$ while $E = \frac{\sum R_j^2}{\sum R_j}$, having the same ingredients, and being $\sum R_j \approx p$, I am not surprised that the two quantities are highly correlated. These computations lead me to a question. In equation (1) the authors correctly state that the weights used must be proportional to the infectious force of each group. However, in all subsequent formulae (e.g. equations (3) and (4)) the force of infection of each group disappears, and all groups seem to be equally weighted. Can the authors explain?

Thanks for this insightful comment as it makes very clear why the MSEs associate well. In Eq (1), we computed how local R_j contribute to the commonly used (global) standard R . This led to the weighting by the infectious force Λ_j . This is an entirely sensible weighting but we found that it can conceal key signals for informing policymaking because of the strong dependence on the Λ_j . For example, if one location has a large but slowly declining incidence and another five locations have small incidence but are all growing, $R < 1$ remains until those five regions dominate. This signal may be too late for effective targeted action. Consequently, we explored if better weightings, from the perspective of earlier signalling of stability, are possible.

We first considered what next generation matrix methods imply, obtaining the max R_j . Our logic was then to bridge between weighted averaging and a max operation. This removed an explicit weighting from Λ_j , which was the source of the problem in our use case i.e., if we want a statistic that better reflects the heterogeneity in local groups but still offers some consensus over those groups for timely decision-making, we need to break the explicit link to Λ_j . $E = X(1)$ did exactly this, replacing weights $\Lambda_j / \sum_i \Lambda_i$ with $R_j / \sum_i R_i$.

However, E still implicitly includes the Λ_j . This follows because the Fisher information of every R_j estimate is proportional to Λ_j , so the resulting uncertainty in E factors in forces of infection. Hence as infections and R_j estimates are inherently linked, Eqs (3)-(4) do not ignore or equally weight infectious forces. Instead, they are weighted differently and implicitly. The link between uncertainty and Λ_j is important because the false positive and false negative signals of stability from R and max R_j occur because they respectively under and over-estimate uncertainty. We have added text near Eq (3) to clarify this implicit dependence.

5) In this manuscript, all estimates are performed separately for the groups, as if the epidemic dynamics were independent. The authors explain that this approach is necessary, since in real time information about contact patterns or mobility will not be readily available. I accept the argument, but an obvious question is whether the E statistics correctly captures the

epidemic trend when the groups are indeed connected. I think that this could be answered by performing stochastic simulations where the groups are connected through some contact matrix, so that the epidemic trend is determined by the maximum eigenvalue of the next-generation matrix (or the equivalent formulation in terms of transfer function); assuming however that surveillance has access only to (noisy) incidence data, one can test whether the E statistics correctly (or, at least, better than alternatives) access the stability threshold.

Thanks for this interesting suggestion. We explore the impact of interactions in responses 2) and 3) to Reviewer 1 above. In 2) we detail and bolster the use-case of E , explaining why by design we focussed on improving transmissibility information under commonly used, non-interacting assumptions. E aims to refine what is currently done during unfolding outbreaks, where contact and mobility data are unreliable, unavailable and changing due to unexpected behaviours. Additionally, locations often want to act swiftly and independently (federated surveillance and control) with minimal computational burden and easily interpreted statistics, all of which we built into E . Importantly, as we only use limited information, we do not expect E to outperform a metric that uses mobility information.

However, we do find, in response 3) to Reviewer 1, that $E=1$ is a good approximation to the epidemic threshold (from transfer function or next generation methods) that perfectly knows interactions, whenever migration is symmetric, local reproduction numbers converge or when one location dominates migration (unidirectional flow). In the last setting the local R_j definition changes, reflecting the complexity of thinking both locally and globally when incorporating interactions. The standard R also still applies in all scenarios and is outperformed by E . We preferred to directly examine mathematical model formulae because of the rich dynamics that can emerge from interactions. Stochastic simulations would limit our ability extract generalised insights and cover such a wide space of interaction probability options.

We address this in a new Results section, explaining when E is useful and acknowledging the importance of interactions. We detail the formulae supporting our claims and their implications in a new Supplement section. We also highlight that in our two empirical studies (Veneto, Italy and Texas, USA) E was able to improve on the current best practice (where interaction information is unavailable) even though in reality some level of interactions would underpin this empirical data. Ultimately, our study is about balancing among descriptive power, speed and assumptions (e.g., maybe ignoring interactions can be better than getting them wrong), which remains an open challenge in epidemic modelling.

6) As a final comment, let me say that, contrary to the authors' remarks, the statistics $\max R_j$ does not seem to perform so badly. In Figure 4, it provides estimates very similar to E ; in Fig. 5, it is consistently above 1 up to about November 15, but this seems quite in agreement with actual trends.

We appreciate this comment and note that $\max R_j$ does have a basis from the transfer function approach and as the limit of the continuum of X estimators. However, Fig 5 shows that it just always remains above 1, which from a policy informative perspective is much too conservative and insensitive to important transmissibility variations. Importantly, $\max R_j$ can be difficult to interpret as it remains above 1, even when total infections are declining, provided there exists

one slightly growing group or some noise. We have balanced our discussion of $\max R_j$ in the text where it is derived by noting that it does embody a key limit as if it is 1, there is no chance of epidemic growth. This may be useful for highly stringent approaches to disease control such as the zero COVID policy Hong Kong adopted prior to the Omicron variant.

Reviewer #3:

The authors investigate cases where the epidemic stability threshold, $R=1$, may lose its conventional meaning. This topic is interesting and potentially valuable. However, the current manuscript suffers from structural and clarity issues that significantly hinder comprehension. Therefore, I do not recommend publication in its current form. A thorough revision is needed before the manuscript can be reconsidered. My specific comments are as follows:

Thanks for the constructive comments that have improved the paper.

1) Manuscript Structure:

The overall structure should be reorganized for clarity. For example, in Fig. 1, the compartment E appears, yet its meaning is not introduced until Eq. (4) later in the text. Such delayed definitions make it difficult for the reader to follow the model formulation. I suggest introducing all variables and compartments before presenting the figures and model equations.

We have included clarifying text before the figure to explain E is another reproduction number that we will derive and use for comparison. Reorganisation (without substantial repetition) was not possible as it is helpful for the reader to understand the standard metrics (e.g., R) as they are introduced. As our goal is to compare metrics, it is then useful for the reader to return to that figure with knowledge of E . We have now inserted text across the Results to help the reader in examining and comparing these metrics. We have also added detailed information on the simulations underlying figures to enhance clarity (see responses 2)-3) to Reviewer 2).

2) Conceptual Clarity in Key Sections:

The arguments in the sections "Reproduction numbers may provide false positive indicators of stability" and "Next-generation matrices may provide false negative indicators of stability" are unclear. These are central claims, yet the manuscript lacks detailed explanation and rigorous justification. Please provide clearer theoretical arguments, illustrative examples where necessary, and a step-by-step explanation to support these claims.

We have now added better and clearer definitions of terms such as false positive and negative indicators of stability, establishing why we use these terms and which parts of the results they describe. We have also inserted text throughout the Results sections to make arguments as clear and mathematically intuitive as possible. As per our responses to Reviewer 2 on the key simulations in the paper, we have now included substantially more detail around the analyses that underpin or illustrate our arguments. These additions aid clarity and reproducibility.

3) Formatting and Presentation:

There are several formatting issues that should be corrected. For instance, all equation numbers should be right aligned for consistency with standard mathematical formatting. A careful proofreading of the entire manuscript is recommended.

We appreciate this suggestion but refrain from making formatting changes at this stage as if eventually successful, the formatting would be controlled by journal guidelines. We have read through the paper thoroughly and made other edits, correcting any typos and other issues.

Bibliography

1. Arregui S, Aleta A, Sanz J, Moreno Y. Projecting social contact matrices to different demographic structures. *PLoS Comput Biol*. 2018;14: e1006638. doi:10.1371/journal.pcbi.1006638
2. Hamilton MA, Knight J, Mishra S. Examining the influence of imbalanced social contact matrices in epidemic models. *Am J Epidemiol*. 2024;193: 339–347. doi:10.1093/aje/kwad185
3. Liu Q-H, Ajelli M, Aleta A, Merler S, Moreno Y, Vespignani A. Measurability of the epidemic reproduction number in data-driven contact networks. *Proc Natl Acad Sci USA*. 2018;115: 12680–12685. doi:10.1073/pnas.1811115115
4. Nouvellet P, Bhatia S, Cori A, Ainslie KEC, Baguelin M, Bhatt S, et al. Reduction in mobility and COVID-19 transmission. *Nat Commun*. 2021;12: 1090. doi:10.1038/s41467-021-21358-2
5. Lyu R, Rosenfeld R, Wilder B. Federated epidemic surveillance. *PLoS Comput Biol*. 2025;21: e1012907. doi:10.1371/journal.pcbi.1012907
6. Mishra S, Scott JA, Laydon DJ, Zhu H, Ferguson NM, Bhatt S, et al. A COVID-19 Model for Local Authorities of the United Kingdom. *Journal of the Royal Statistical Society Series A: Statistics in Society*. 2022;185: S86–S95. doi:10.1111/rssa.12988
7. Parag KV, Cowling BJ, Donnelly CA. Deciphering early-warning signals of SARS-CoV-2 elimination and resurgence from limited data at multiple scales. *J R Soc Interface*. 2021;18: 20210569. doi:10.1098/rsif.2021.0569
8. Bouros I, Thompson RN, Gavaghan DJ, Lambert B. The time-dependent reproduction number for epidemics in heterogeneous populations. *J R Soc Interface*. 2025;22: 20250095. doi:10.1098/rsif.2025.0095
9. Trevisin C, Bertuzzo E, Pasetto D, Mari L, Miccoli S, Casagrandi R, et al. Spatially explicit effective reproduction numbers from incidence and mobility data. *Proc Natl Acad Sci USA*. 2023;120: e2219816120. doi:10.1073/pnas.2219816120
10. Roy M, Clapham HE, Mishra S. Incorporating human mobility to enhance epidemic response and estimate real-time reproduction numbers. *PLoS Comput Biol*. 2025;21: e1013642. doi:10.1371/journal.pcbi.1013642
11. Birello P, Re Fiorentin M, Wang B, Colizza V, Valdano E. Estimates of the reproduction ratio from epidemic surveillance may be biased in spatially structured populations. *Nat Phys*. 2024; doi:10.1038/s41567-024-02471-7
12. Parag KV. Improved estimation of time-varying reproduction numbers at low case incidence and between epidemic waves. *PLoS Comput Biol*. 2021;17: e1009347. doi:10.1371/journal.pcbi.1009347

Responses to Reviewers: COMMSPHYS-25-1582A

We thank all reviewers and the editor for their constructive and insightful comments. We list our responses to the last comments from the reviewers.

Reviewer #1

I approve all changes made to the manuscript by the authors in response to the reviewer comments. I have seen one small typo: on page 2, "as both a prospective or retrospective", replace or with and.

Thanks for spotting this, completed.

Reviewer #2

I believe that the revised version successfully addresses the reviewers' comments and is acceptable for publication. I only have two remarks:

- In the legend of Fig. 3 the authors state "We identify three regions of apparent overall stability (black, dashed)". I think that the word "region" is confusing, both because "region" is used in the previous sentence to denote a spatial unit, and (more important) "region" generally denotes an extended time length (that one does not see in the Figure), while here I would say "three times of apparent..." or "three time points of apparent..."

We have made these changes in the main text.

- In the new paragraph (which I appreciated) "Epidemic thresholds outside of real-time, data-limited settings", the authors state "whenever migration is symmetric... E performs well when compared to T". However, looking in the Supplement at case 2, one reads " $T=1$ requires $(\delta+\epsilon)(1-\rho)\approx 0$ while $E=1$ requires $0.5(\delta+\epsilon)\approx 0$ ". If ρ is close to 1 (as I would expect: most contacts in one's own group), say $\rho=0.9$, the threshold for E is off almost by a factor 2. I would not say it performs well in such a case. Also it seems that the use of E works best when $\rho=1/2$, meaning that essentially one has a well-mixed population with two types differing in transmissibility, but not very well when transmission between groups is small. I believe that this observation deserves a comment.

Thanks for this suggestion, we have made this clarification as suggested.

Reviewer #3

The authors have addressed all my questions. I suggest it for publications.

Thanks.

$$R_t^{(k)} = R_t \frac{\left(\sum_{i=1}^N \Lambda_t^{(i)} \right) \left(\sum_{j=1}^N C_t^{(jk)} \right)}{\sum_{i=1}^N \sum_{j=1}^N C_t^{(ji)} \Lambda_t^{(i)}}.$$

Comments on the manuscript “When is the $R = 1$ epidemic stability threshold meaningful?”– by Parag et al.

The main message of the manuscript under review is that, when a population is split into different groups, and separate incidence data are available for these groups, a summary statistics, named E , that averages in a suitable way the effective reproduction numbers R_j computed for each group, can be an appropriate indicator for guiding public health policy in real time, much better than the effective reproduction number R computed for the whole population.

In order to support the use of E , the authors consider the example of the data of Covid-19 in the autumn 2020 in the Veneto region of Italy, as well as some simulations that appear to have been structured to yield incidence data somewhat reminiscent of those of the Veneto region.

I think this makes an interesting contribution to the field. However, I also think that the manuscript needs be improved before publication.

A first observation, somewhat marginal, is that the authors introduce the formalism of transfer functions, in the Results (6th page of the text) and later in the Methods, without any obvious connection to the rest of the manuscript, except to state the well-known result that in theory the stability region requires $\max R_i < 1$.

The main problem in the manuscript lies, in my opinion, in the lack of details about the simulations performed. Furthermore, I would have liked to see the effectiveness of E in model-based simulations where the ground truth is known and clear.

From the text one can only guess how the simulations shown in Fig. 3 have been obtained. It is true that the code is available in Github, so one could read the code to understand it, but I would expect to get the main elements of the simulations from the text. I gather that the authors have used (slowly) time-varying reproduction numbers, varying asynchronously in the two groups, and that incidence data have been simulated drawing random variables (Poisson?) with mean given by the renewal formula. Possibly the assumed values for the reproduction numbers are close to the estimates, but definitely we are missing most details of the simulations.

Similarly for Fig.2 they state that they sampled 1,000 weighing from the Dirichlet distribution (presumably the mean a symmetric Dirichlet distribution, but do not specify its parameter); I understand that in this way weights w_i (between 0 and 1 that sum up to 1) are obtained. However, the authors need R_i whose average is (approximately?) 1; perhaps, they obtain $E(R_i) = pw_i$, and then simulate 5,000 samples of R_i (with an un-

known distribution), but this is only my guess. The authors should provide some details. Concerning Figure 2, I must add that MSE of the $R_j = \sum R_j^2 - 2 \sum R_j + p$ while $E = \frac{\sum R_j^2}{\sum R_j}$; having the same ingredients, and being $\sum R_j \approx p$, I am not surprised that the two quantities are highly correlated.

These computations lead me to a question. In equation (1) the authors correctly state that the weights used must be proportional to the infectious force of each group. However, in all subsequent formulae (e.g. equations (3) and (4)) the force of infection of each group disappear, and all groups seem to be equally weighted. Can the authors explain?

In this manuscript, all estimates are performed separately for the groups, as if the epidemic dynamics were independent. The authors explain that this approach is necessary, since in real time information about contact patterns or mobility will not be readily available. I accept the argument, but an obvious question is whether the E statistics correctly captures the epidemic trend when the groups are indeed connected. I think that this could be answered by performing stochastic simulations where the groups are connected through some contact matrix, so that the epidemic trend is determined by the maximum eigenvalue of the next-generation matrix (or the equivalent formulation in terms of transfer function); assuming however that surveillance has access only to (noisy) incidence data, one can test whether the E statistics correctly (or, at least, better than alternatives) access the stability threshold.

As a final comment, let me say that, contrary to the authors' remarks, the statistics $\max R_j$ does not seem to perform so badly. In Figure 4, it provides estimates very similar to E ; in Fig. 5, it is consistently above 1 up to about November 15, but this seems quite in agreement with actual trends.